# Chicken Erythrocyte: Epigenomic Regulation of Gene Activity

**DOI:** 10.3390/ijms24098287

**Published:** 2023-05-05

**Authors:** Tasnim H. Beacon, James R. Davie

**Affiliations:** Department of Biochemistry and Medical Genetics, University of Manitoba, Winnipeg, MB R3E 0J9, Canada

**Keywords:** chicken erythrocyte, epigenetics, histone (H4R3me2a) and DNA modifications

## Abstract

The chicken genome is one-third the size of the human genome and has a similarity of sixty percent when it comes to gene content. Harboring similar genome sequences, chickens’ gene arrangement is closer to the human genomic organization than it is to rodents. Chickens have been used as model organisms to study evolution, epigenome, and diseases. The chicken nucleated erythrocyte’s physiological function is to carry oxygen to the tissues and remove carbon dioxide. The erythrocyte also supports the innate immune response in protecting the chicken from pathogens. Among the highly studied aspects in the field of epigenetics are modifications of DNA, histones, and their variants. In understanding the organization of transcriptionally active chromatin, studies on the chicken nucleated erythrocyte have been important. Through the application of a variety of epigenomic approaches, we and others have determined the chromatin structure of expressed/poised genes involved in the physiological functions of the erythrocyte. As the chicken erythrocyte has a nucleus and is readily isolated from the animal, the chicken erythrocyte epigenome has been studied as a biomarker of an animal’s long-term exposure to stress. In this review, epigenomic features that allow erythroid gene expression in a highly repressive chromatin background are presented.

## 1. Introduction

Chicken mature erythrocytes are nucleated cells. Chicken (*Gallus gallus domesticus*), a subspecies of the red jungle fowl (Gallus gallus), has a karyotype consisting of 78 chromosomes. Chromosomes 1–10, Z, and W are classified as macrochromosomes and 11–38 are microchromosomes. The domesticated chicken (poultry) diverged from their ancestor more than 8000 years ago [1]. Their genetic diversion is a result of environmental adaptation or genetic modification to improve the breed, egg, and meat quality. With about the same number of genes as the human, the chicken genome consists of about 60% of at least one human gene orthologue. Furthermore, the nucleated chicken erythrocyte would be an important model to study apoptosis and linked pathways, as evidenced in human erythrocytes [2]. The chicken and human genomes share long blocks of conserved synteny, making the chicken genome an ideal model system for studying vertebrate gene expression regulation. Epigenetic mechanisms, the processes that regulate gene activity and expression that are not dependent on alterations in DNA sequence, include histone post-translational modification (PTMs) and DNA modifications. These epigenetic processes are similar in chicken and human cells.

The key physiological function of the erythrocyte is to carry oxygen to the tissues. Another role of the chicken erythrocyte is participating in the innate immune response. In this review, we will discuss studies on the epigenomic organization of the chicken erythroid cell and the chromatin structure of key genes participating in oxygen-carrying and innate immunity functions.

## 2. Chromatin Structure of the Chicken Erythrocyte

As mature erythrocytes have low transcriptional activity, our studies have often been done with transcriptionally active polychromatic erythrocytes isolated from birds that were made anemic by injection with phenylhydrazine. Another chicken cell of interest in our studies is the 6C2 cell. These cells are avian erythroblastosis virus-transformed cells from the bone marrow that are arrested at the colony-forming unit stage [3]. We studied the structure of erythroid chromatin using mature erythrocytes, polychromatic erythrocytes, and to a lesser extent 6C2 cells.

The chicken’s mature erythrocyte retains its nucleus. The chromatin structure of the mature erythrocyte is very compact due to a high content of linker histones, H1 and H5, which are present in the ratio of 0.4:0.9 per nucleosome. The chicken erythrocyte chromatin is organized as a compact 30 nm fiber, which is rare in other eukaryotic cells [4] (Figure 1). Although thought to be transcriptionally silent, the mature erythrocyte is capable of transcribing genes [5,6]. The transcriptionally active polychromatic erythrocyte genes are expressed in a background of highly condensed chromatin, which presents a challenge to the expressed genes in maintaining a decondensed chromatin state.

The basic unit of chromatin is the nucleosome. The nucleosome is composed of the core histones H2A, H2B, H3, and H4. Except for H4, the core histones have variants [7]. For example, histone H3 has several variants, including H3.1, H3.2, and H3.3. H3.1 and H3.2 are synthesized during the S-phase of the cell cycle and are referred to as replication-dependent, while H3.3 is synthesized throughout the cell cycle and is classified as replication-independent. Chicken erythrocytes express H3.2 and H3.3, but not H3.1. Mature and polychromatic erythrocytes have ceased replication and are in the G0 phase of the cell cycle.

The transcriptionally active/poised/euchromatin state is decondensed chromatin associated with histone PTMs that are referred to as active marks (Figure 1 and Figure 2). Active marks include acetylated histones, H3 trimethylated at lysine 4 (H3K4me3), H3 acetylated at lysine 27 (H3K27ac), and H3 trimethylated at lysine 36. The repressed chromatin state is associated with repressive marks (H3 trimethylated at lysine 9; H3 trimethylated at lysine 27) (Figure 1 and Figure 2). The highly acetylated state of transcriptionally active/poised chromatin prevents H1/H5-induced chromatin condensation (30 nm fiber) and insolubility at 150 mM NaCl [8]. Incubating chicken polychromatic erythrocytes in the absence of a histone deacetylase inhibitor (sodium butyrate) resulted in the deacetylation of transcriptionally active/poised chromatin. In the deacetylated state, the transcriptionally active/poised chromatin fragments underwent H1/H5-induced condensation and salt-insolubility, demonstrating the importance of histone acetylation in maintaining the decondensed state of transcriptionally active/poised chromatin. These studies used labeled DNA probes to transcription active/poised (e.g., *HBBA*, *H1FO*) and repressed (*VTG1*) gene sequences in Southern blot experiments, PCR with primers to active or repressed DNA sequences, and next-generation DNA sequencing to determine the structure (e.g., DNase I sensitivity) and salt solubility properties of active/poised and repressed gene chromatin [9,10,11].

Active and repressive marks define two states of chromatin called compartment A (active marks) and compartment B (repressive marks), respectively [12]. Another feature of the chromatin structure in vertebrate cells is the organization of the chromatin into topologically associated domains (TADs) [13], which are chromatin loops formed by the action of the cohesin complex and CTCF (Figure 2). However, chicken polychromatic and mature erythrocyte chromatin are not organized into TADs [14,15]. This was a surprising finding because chicken erythrocytes express CTCF; however, the erythrocyte CTCF is not associated with chromatin and is present in large nuclear matrix-associated bodies [16], which explains the lack of TADs in chicken erythroid cells.

## 3. Chicken Erythrocyte Promoters

A prominent DNA sequence feature seen in vertebrate genomes is the CpG island, which is normally not methylated (5 methylcytosine). A CpG island is defined as having an average % G + C content higher than 50%, an observed CpG versus expected CpG ratio greater or equal to 0.6, and a length greater than 200 base pairs [17]. Chicken promoters can be classified into two categories: promoters with long CpG islands (CGIs) (>800 bp) promoters, which is the case for the top 10% of the genes in terms of CGI length and biological processes such as morphogenesis, development, transcription processes, and biosynthetic processes [18]. “No CGI” refers to promoters without CGI and typically are associated with biological processes such as the immune system and defense mechanisms. Another feature of chicken promoters is a potential G-quadruplex sequence which is prominent in genes associated with development and morphogenesis. To better understand the evolution of promoter regions in vertebrates, researchers have compared the abundance and distribution patterns of sequence motifs in these regions [18]. Activation of interferon-induced and IRF3-dependent primary response genes with promoters without CGIs requires the chromatin remodeler, SWI/SNF-remodeling complex [19,20]. Although the chicken genome has a much smaller number of repetitive elements in comparison to mammals, the short tandem repeat motif frequency in the chicken promoter was more comparable to the mammalian promoters. Scientists interpret the findings to be linked to conservation through promoter evolution and insightful regarding the structure of avian promoters [18].

Typically, the promoter of a transcriptionally active gene is marked by the presence of H3K4me3, H3K9ac (H3 acetylated at lysine 9), and H3K27ac. H3 symmetrically dimethylated at arginine 2 (H3R2me2s) is usually localized at the promoter region with the active mark H3K4me3 and is known to guide binding factors to recognize the promoter mark H3K4me3 [21,22]. The presence of H3K4me3, H3K4me2, and H3K27me3 at the same time marks a ‘poised promoter’ or gene that has its fate yet to be decided. H3R2me2s is also known to mark regions enriched with acetylated H3/H4 at poised promoters. The function of H3R2me2s is known to keep genes poised until transcriptional activation or differentiation [22].

Promoters, enhancers, and locus control regions are associated with transcription factors and are devoid of nucleosomes (Figure 2). Several methods have been applied to determine the location of these nucleosome-free regions, including DNase 1 hypersensitivity Sequencing, ATAC Seq, and FAIRE (formaldehyde-assisted isolation of regulatory elements using sequencing) Seq, which is the method that we used. The transcription factors residing in the nucleosome-free regions can directly or indirectly recruit chromatin-modifying enzymes and modify the core histones of nucleosomes residing next to the nucleosome-free region. Our analysis showed that FAIRE Seq readily identified promoter regions but failed to map out the enhancers as efficiently [23]. The enhancer positions however were determined by FAIRE PCR, which is the analyses of the DNA by PCR instead of by next-generation sequencing. Typically, the peaks identified by FAIRE Seq were localized to the 5′ regions of the gene body.

## 4. DNA Methylation

Another level of control of the chromatin state is offered through DNA methylation catalyzed by the DNA methyltransferase enzymes (DNMTs). The de novo methyltransferases (DNMT3a and DNMT3b) catalyze methylation by the addition of a methyl group onto the fifth carbon of cytosine residues in the DNA (5mC) position, whereas DNMT1 functions to maintain DNA methylation through the addition of methyl groups during replication. The major function of these enzymes is to establish and maintain heterochromatinization and gene silencing, and promote transcriptional repression. In contrast to the previous knowledge of DNA methylation, the current investigation of methylome profiles shows DNA methylation within the gene bodies, which correlates positively with gene expression. Vertebrate-expressed genes are typically unmethylated at the 5′ promoter region and methylated throughout the gene body [24]. β-globin genes of chicken erythrocytes were found to be hypomethylated when compared with other chicken somatic tissues. This study was possible only because of the unique gene expression of the chicken model system in which the globin genes are differentially expressed in embryo and adult erythrocytes. For example, Haigh et al. found that the *HBZ* (αD) genes in the embryonic and adult erythrocytes are hypomethylated which is not the case with brain and sperm *HBZ* genes [25]. These studies on DNA methylation of chicken mature and polychromatic erythrocytes were done using methylation-sensitive (HpaII) and insensitive (Msp1) restriction endonucleases.

DNA methylation in the gene body is conserved and maintained across various species by both DNMT1 and the methylation binding co-factor ubiquitin-like containing PHD and ring finger domains (UHRF1) [24]. The gene-silencing function of DNA methylation is achieved by the recruitment of histone deacetylase (HDAC) complexes that have methyl-DNA binding motifs, and through relationships between lysine methyltransferases (Suppressor of Variegation 3-9 Homolog 1/2, SUV39h ½, G9) and DNA methylation [26]. It was found that DNMT1 also serves as a key player in recognizing evolutionarily divergent target sequences [27].

DNA methylation at the promoter may prevent the binding of transcription factors as well as attract readers to bring in protein complexes that maintain transcriptional inactivity. Although very few studies about the methylation pattern have been reported, chicken has been used to find the function of the DNA demethylase ten-eleven translocation (TET). Five mC is oxidized by the TET family of enzymes, TET1, TET2, and TET3 which convert 5mC to 5-hydroxymethylcytosine (5hmC). In a study of the TET enzyme, one group cloned the chicken *TET* genes and found their regulatory function over the adult β globin genes [28]. TET was seen to cause demethylation of the promoters of the *HBBA* (βᴬ) globin gene in erythroid cells. The same group observed decreased expression of the *HBBA* globin gene during *TET1* knockdown experiment in chicken erythroid progenitor cells [28]. Using the chicken B cell lymphoma cell line DT40, investigators found that a high level of DNA methylation was maintained and regulated by inhibiting access to TET enzymes, which, in turn, may be regulated by repressive histone marks [27].

As the chicken mature erythrocyte is nucleated and is readily accessible from the living animal, researchers have explored the impact of long-term stress on the erythrocyte DNA methylome [29,30]. Pértille et al. applied an approach called GBS-MeDIP (genotype-by-sequencing (GBS) and methylated-DNA-immunoprecipitation (MeDIP)) [31] to identify differentially methylated DNA regions (DMRs) in female Dekalb white chicken erythrocytes. This approach determined the methylation status of about 7.6% (810,186 sites) of all CpGs in the chicken genome [29]. The goal of the 2017 study was to determine the impact of stress on female chickens that were reared under different conditions (open aviaries versus cages). DMRs were observed on all chromosomes except for chromosomes 3, 9, 14, 18, 23, 32, and W. The DMRs were common in regulatory regions. Analyses of the genes with DMRs by Consensus PathDB found that these genes were involved in signal transduction pathways (MAPK signaling, G protein activation), and when analyzed using Reactome, the differentially methylated genes had functions in the immune system. In the more recent 2020 study, Pértille et al. studied the impact of chronic stress (social isolation) on the DNA methylome of erythrocytes isolated from male White Leghorn chickens from two different locations (Brazil and Sweden) [30]. The erythrocyte DMRs were often located around a gene’s transcription start site, suggesting that the methylation event may impact transcription factor binding (e.g., RELA). The DMRs for birds in Brazil and Sweden were pronounced in microchromosomes and ChrZ. The results of these studies clearly show that long-term stressful conditions impact the erythrocyte DNA methylome which likely affects other epigenetic processes and gene expression patterns. With red blood cells being easily accessible, the chicken erythrocyte model system is ideal in studies exploring the impact of environmental conditions on the epigenome in live animals.

## 5. Chromatin Fractionation: A Powerful Method to Study Chromatin Composition of Expressed and Repressed Chromatin

A major advantage to using the chicken erythrocyte to study chromatin structure is the features of these nucleated cells lend themselves to the fractionation and isolation of transcriptionally active/poised oligonucleosomes and polynucleosomes [12]. The transcriptionally poised/active chromatin (compartment A) of chicken mature and polychromatic erythrocytes have highly acetylated histones, and the erythroid chromatin has high levels of the linker histones H1 and H5, which together make the isolation of transcriptionally active chromatin possible [12]. The first steps in the chromatin fractionation protocol include micrococcal nuclease digestion of chromatin and lysis of the nuclei. Under these conditions, chromatin fibers decondense and are soluble. Following centrifugation, most chromatin fragments (active/poised and repressed DNA) are present in the supernatant (fraction S_E_), while the pellet (fraction P_E_) has some chromatin (both transcriptionally active and repressed genes) that is bound to the insoluble nuclear material which includes the nuclear lamina and nuclear matrix. The addition of NaCl (to 150 mM) to the S_E_ fraction renders most chromatin fragments insoluble. The insoluble chromatin fragments form higher-order chromatin structures such as the 30 nm fiber. The insoluble chromatin fragments are collected by centrifugation, yielding fraction P_150_. The supernatant has salt-soluble chromatin fragments (polynucleosomes, oligonucleosomes, and mononucleosomes) (fraction S_150_). The salt-soluble polynucleosomes and oligonucleosomes are enriched in active/poised DNA sequences. The mononucleosomes are derived from active/poised and repressed chromatin. Gel exclusion chromatography is applied to separate the longer chromatin fragments (fractions F_1–3_) from the mononucleosomes (fraction F_4_). As shown in Figure 3, transcriptionally active/poised chromatin is present in fractions S_150_, F_1–3_, and P_E_, while repressed chromatin is in fractions P_150_ and P_E_. Next-generation DNA sequencing of the F_1_ chromatin fraction identified the salt-soluble chromatin, DNase I sensitive domains in the chicken polychromatic erythrocyte [11]. Details of the chromatin fractionation protocol are found in [9,32].

Polynucleosomes present in the transcriptionally active/poised chromatin fraction F_1_ have canonical nucleosomes and atypical nucleosomes that are described as U-shaped nucleosomes [33]. The U-shaped nucleosome is formed during transcriptional elongation [34] and will remain in this altered structural state until the U-shaped nucleosome is deacetylated [35]. The presence of these U-shaped nucleosomes in the polynucleosomes suggests that these atypical nucleosomes are not overly sensitive to micrococcal nuclease digestion. The structural alteration in the U-shaped nucleosome results in the exposure of the normally buried H3 sulfhydryl cysteine (Cys 110). Vincent Allfrey and colleagues exploited this feature of the U-shaped nucleosome to capture these structures on a mercury column, allowing for the characterization of the U-shaped nucleosome [36].

## 6. Chicken Erythrocyte Core Histone PTMs

### 6.1. Metabolism

Metabolism and epigenetic processes are intertwined. Metabolites are required to drive the activity of the enzymes catalyzing epigenetic processes such as histone and DNA modifications and chromatin remodeling [37,38,39]. For example, acetyl CoA is required for the lysine acetyltransferases, S-adenosyl-L-methionine is required for lysine and arginine methyltransferases and DNMTs, and ATP for chromatin remodelers and kinases.

### 6.2. Histone H2A and H2B PTMs

Chicken polychromatic erythrocyte histones H2A and H2B are acetylated and ubiquitinated, with H2B having more acetylated sites than H2A. The H2A variant, H2A.Z, is also acetylated. H2B is the most rapidly acetylated/deacetylated core histone in polychromatic erythrocytes [40]. In chicken polychromatic erythrocytes, ubiquitinated H2A (uH2A) is associated with both expressed and repressed chromatin, while ubiquitinated H2B (uH2B) is present mainly with expressed chromatin (Figure 1). Both uH2A and uH2B are present together in a nucleosome and enriched in nucleosomes of the transcriptionally active/poised chromatin fraction F_1_ [41]. Ubiquitinated H2A and H2B increased the lability of nucleosomes. In contrast, the interaction of the H2A.Z with the (H3-H4)_2_ tetramer and/or nucleosomal DNA was stronger than that of histone H2A [41]. For reasons yet to be explained, uH2B levels in fraction P_E_, which contains most of the transcribed DNA sequences, were very low. Interestingly, the chromatin-associated newly synthesized H2A and H2B were prominently ubiquitinated. In this study, chicken polychromatic erythrocytes were labeled with L-[4,5-^3^H] lysine for one hour to label newly synthesized histones [42]. The labeled histones from unfractionated chromatin and chromatin fractions (S_E_, S_150_, P_150_, P_E_) were resolved by two-dimensional electrophoresis (acetic acid-urea-Triton X-100 (AUT) polyacrylamide gels to SDS polyacrylamide gel electrophoresis (PAGE)) followed by fluorography. AUT-PAGE resolves histones by size, charge, and hydrophobicity [43]. Ubiquitinated H2A and uH2B have distinctive positions on the one-dimensional AUT and two-dimensional gel patterns. The fluorograms showed that uH2A and uH2B in the transcriptionally active/poised-enriched chromatin fraction F_1_ were labeled at levels comparable to or greater than the parent histone. The genomic location of uH2A and uH2B has not yet been determined in chicken erythrocytes, which would provide insights as to where the newly synthesized histones are going in the genome.

### 6.3. Histone H3 PTMs

H3 has two variants in chicken erythroid cells, H3.2 and H3.3. Both H3 variants are acetylated (K27ac), methylated (K4me1, K4me2, K9me2, K4me3, R2me2a, R2me2s), and phosphorylated (S10, S28). The H3 of transcriptionally active chromatin (fraction F_1_) is acetylated (including K27ac), methylated (K4me1-3, R2me2s), and phosphorylated (S28ph). Repressed chromatin (fraction P_150_) has H3K9me2, H3S10ph, and H3R2me2a (Figure 1). Histone immunoprecipitation/immunoblotting studies showed that an H3 modified at R2me2s can be also modified at K4me1, K4me3, and K27ac [44]. The replacement histone H3.3 is preferentially phosphorylated at S28, methylated at K4 (K4me3), and methylated at R2 (R2me2s) [45]. Studies are ongoing to determine which other H3 modifications are associated with expressed and repressed erythroid chromatin.

### 6.4. Histone H4 PTMs

In transcriptionally active chromatin, H4 is dynamically acetylated and methylated at R3 (R3me2a) [45,46,47] (Figure 1). Mass spectrometry analyses show that most H4 is dimethylated at K20 [48].

## 7. Histone Acetylation, Chromatin Solubility, and the DNase I Sensitive Chromatin Domains

Early studies on the characterization of chicken erythrocyte histone PTMs were done by electrophoretic separation of radiolabeled (e.g., ^3^H acetate) histones by acetic-acid-urea-Triton X100 (AUT) polyacrylamide gel electrophoresis (PAGE) [43]. With the availability of antibodies to specific histone PTMs, histones resolved by AUT PAGE were analyzed by immunoblotting.

In chickens, mature and polychromatic erythrocytes only a small proportion (about 2%) of the total genome participates in dynamic histone acetylation. The studies done by Zhang and Nelson demonstrate that core histones associated with transcriptionally active chromatin are rapidly acetylated and deacetylated [40,49]. In polychromatic erythrocytes, H4 is acetylated at a rate with a half-life of 12 min, while in mature erythrocytes, H4 is acetylated at two rates (t½ of 12 min and 300 min) [40]. The highly acetylated histone (e.g., tetra acetylated H4) is rapidly deacetylated at a rate of 5 min in mature and polychromatic erythrocytes [49]. The most rapidly acetylated and deacetylated histone was H2B.

The lysine acetyltransferase (KAT) responsible for acetylating several lysine sites on H3, H4, and H2B is CBP/p300 [50]; both enzymes are expressed in chicken polychromatic erythrocytes. In contrast to the histone deacetylases (HDACs) [51,52], there has not been much attention to characterizing the chicken erythrocyte KATs [53]. In the chicken mature and polychromatic erythrocyte nucleus, most HDAC and KAT activities are associated with the nuclear matrix [54,55]. One interesting finding from the mass spectrometry data showed phosphorylated HDAC2 at promoters forms multiprotein complexes while unmodified HDAC2 had a bias to bind to RNA splicing proteins [51].

Epigenetic research has a history of studying the chromatin state using the chicken organism [56]. Like the human genome, the chicken genome has transcriptionally active and repressed chromatin states. Transcriptionally active/poised chromatin of mature and polychromatic erythroid cells has a more accessible structure than the bulk of highly condensed repressed gene chromatin as shown by having an increased sensitivity to DNase I digestion (Figure 1). The chromatin of genes in mature erythrocytes that are DNase I sensitive is referred to as being in a poised state.

The majority of chicken erythrocyte chromatin is rendered insoluble at physiological ionic strength and is organized as a 30 nm chromatin fiber that is rare in other eucaryotes. In contrast, the dynamically acetylated transcriptionally active/poised chromatin is soluble in 150 mM NaCl (Figure 1). The DNase I sensitivity of the chromatin of chicken erythroid genes is proportional to the gene chromatin’s 150 mM NaCl solubility [10]. The DNase I sensitivity of expressed/poised chromatin is not limited to the gene but may stretch many kilobases on either side of the gene, defining a DNase I sensitive domain which is associated with acetylated histones [57,58]. Although we know that histone acetylation is a critical player in maintaining the salt-soluble DNase I sensitive chromatin structure, we do not know which acetylated histones and acetylation sites are responsible for this decondensed chromatin structure.

The DNase I sensitivity of transcriptionally active/poised chromatin has been exploited to learn the location of expressed/poised genes in the erythrocyte nucleus. Labeling of the DNase I sensitive regions was done by in situ nuclear nick-translation with the DNase I nicks being labeled with biotinyl UTP [59]. The labeled DNA was located at the interchromatin channels which are at the boundaries of the condensed chromatin masses. Indirect immunolocalization of histone deacetylase 2, which is bound to expressed/poised chromatin, also provided evidence that expressed/poised genes are in the interchromatin channels [12].

## 8. Chicken Erythrocyte Histone Genes and Variants

Chicken mature and polychromatic erythrocytes are arrested in the G0 phase of the cell cycle. Transcripts for the four core histones are present in the chicken polychromatic erythrocytes. There are low levels of transcripts for the replication-dependent histones H1 (*HIST1H1C*), H2A (*HIST2H2AC_dup2*), H2B (*HIST1H2BO*, *H2B-V*), and H4 (*H4*, *H4-VII*). In contrast, the expression of several of the replication-independent histone genes (*H3F3B*, *H2AFZ*, *H1F0*) was high. However, it is newly synthesized H2A and H2B that are incorporated into chromatin, with most of these going into compartment A transcriptionally active chromatin [42]. There is much less newly synthesized H3 and H4 incorporated into chromatin, with newly synthesized H3.3 (replication-independent) being the preferential H3 histone variant present in transcribed chromatin. The insertion of newly synthesized core histones into transcriptionally active chromatin was not impacted by the inhibition of transcription. The results of this study are consistent with our observations that transcription-active nucleosomes have a labile structure [37].

*Histone H1F0:* The most prominent chromatin-associated newly synthesized histone in polychromatic erythrocytes was histone H5 which is encoded by the *H1F0* gene [42]. Newly synthesized H5 was associated with both compartment A and B chromatin. Other than the globin genes, the erythroid regulation of the *H1F0* gene expression has been extensively characterized and provides an example of regulatory elements and factors involved in erythroid gene expression. The gene has three enhancers (two upstream and one downstream). Presumably, the enhancers interact with the promoters as shown in Figure 2 to drive the transcription of this gene. We used ligation-mediated PCR in situ footprinting to map the transcription factor binding sites in the *H1F0* promoter and enhancers in chicken polychromatic erythrocytes [60]. Transcription factors Sp1 (or Sp3), upstream promoter element-binding protein, and CACCC (EKLF) were bound to the promoter. The downstream enhancer was associated with transcription factors Sp1/3, AP2, GATA1, and NF1. The promoter and downstream enhancer are also associated with several of these transcription factors in mature erythrocytes. The mature erythrocyte *H1F0* gene is in a poised chromatin state and shows transcriptional activity in permeabilized cells [6]. The *H1F0* gene is associated with a CpG island that spans the entire gene [23]. In polychromatic erythrocytes, the *H1F0* gene is in a broad F_1_ Seq domain, indicative of the solubility of this region at physiological ionic strength. The *H1F0* gene body and flanking regions that have the promoter and enhancers have extensive FAIRE Seq peaks, showing that this region has nucleosome-free regions. Adjacent to the FAIRE Seq peaks are nucleosomes modified at H3 (R2me2s, K4me3, K27ac) and H4R3me2a. In 6C2 cells, the *H1F0* gene promoter region, enhancer, and gene body are associated with nucleosomes containing H3.3 and H2A.Z, which destabilize nucleosome structure [61]. Together these studies show that the *H1F0* gene has an unstable chromatin structure in mature, polychromatic, and 6C2 cells. An important point of this research on the chromatin structure of the *H1F0* gene is that in polychromatic erythrocytes the *H1F0* chromatin is in a state that is soluble at physiological ionic strength while neighboring chromatin regions are not. This salt-soluble chromatin state would readily present the *H1F0* gene and regulatory regions to transcription factors, chromatin-modifying enzymes, and the transcription machinery.

## 9. Chicken Histone PTMs, Nucleosome-Free Regions, and Genomic Mapping

We characterized transcribed genes within the euchromatin/compartment A of chicken polychromatic erythrocytes by profiling several histone PTMs at the promoter, enhancer, and gene body of individual genes. For this purpose, we combined F_1_ Seq, chromatin immunoprecipitation (ChIP) Seq, and RNA Seq with FAIRE Seq to build an epigenomic map. Histone PTMs were located in the chicken erythroid chromatin by using ChIP Seq, which involves immunoprecipitation of fragmented formaldehyde cross-linked chromatin with antibodies specific to a histone PTM. The transcriptome of the erythroid cell is determined by RNA Seq, which is the sequencing of cellular RNA. FAIRE Seq as we presented previously maps genomic regions that are nucleosome-free. We used the *Gallus gallus* galGal6 assembly provided by the Genome Reference Consortium GRCg6a for all the sequence alignments. Sequenced data from each technique were aligned to map out the histone PTMs. To sum up our results with chicken polychromatic erythrocytes, H3K4me3, H3K27ac, H3R2me2s, and H4R3me2a locate at the 5′ end of the expressed gene including the promoter region (Figure 2). Enhancers and locus control regions, which are like super-enhancers, which are a group of enhancers in close genomic proximity, have H3K27ac, H3R2me2s, and H4R3me2a. To date, we are the only lab to map both H3R2me2s and H4R3me2a [45]. Our study showed that these two marks are often together and locate in regulatory regions of the chicken erythroid genome. H3S28ph is located at promoters of expressed genes in polychromatic but not mature erythrocytes [62]. The genomic location of H3S28ph was determined by ChIP qPCR, which analyzes the immunoprecipitated DNA using PCR rather than next-generational DNA sequencing, but not yet by ChIP Seq.

## 10. Broad Histone PTM Domains and Chromatin Remodeling by CHD1

Most expressed genes have H3K4me3 located in a narrow region at the 5′ end of the gene (narrow domains); however, a small subset of genes has a broad H3K4me3 domain that extensively covers the coding region. Genes configured with the broad domain are involved in cell identity [37]. In addition to H3K4me3, these chromatin domains are associated with multiple modified histones (H3K27ac, H3R2me2s, H4R3me2a) and have a labile structure. We proposed that the highly modified nucleosomes associated with the gene bodies were going through cycles of dissolution and reassembly [63]. Many factors promote the disassembly/displacement of nucleosomes to clear the passage for RNA polymerase II. They also help to reassemble nucleosomes after the polymerase has passed. We speculated the key player behind the unstable nature could be the ATP-dependent chromatin remodeler CHD1 (chromodomain helicase DNA binding protein 1) as it is directly involved in nucleosome assembling, remodeling, sliding, and promotes their regular spacing [64,65] and was identified as a predictor of the broad H3K4me3/H3K27ac domain [66]. For further information on the mechanism of CHD1 remodeling see [67,68]. CHD1 is one of the members of a highly conserved chromatin remodeling enzyme family found in all eukaryotes [65]. The RNA polymerase II-associated PAF1 complex plays a key role in the recruitment of CHD1 to actively transcribed genes [69,70] and the chromodomain of CHD1 binds to H3K4me3 [71]. CHD1 has a role in the exchange of conical histones with histone variants (H2A.Z, H3.3) [64,72]. We postulate that CHD1 may have a role in incorporating H3.3 into erythroid-transcribed chromatin. Studies with chicken erythroid cells show that nucleosomes with the histone variants H3.3 and H2A.Z are unstable [73]. Adding to this we reported that H3.3 may be modified at several sites (R2me2a, K4me3, S28ph). Together, the highly modified nucleosomes of broad histone PTM domains and CHD1 may contribute to the instability of the nucleosomes of genes with this chromatin configuration. It is acknowledged, however, that there are other CHD chromatin remodelers that may be involved in nucleosome remodeling of broad histone PTM domains [74].

The chicken *CHD1* gene is located on the sex chromosomes Z (*CHD1Z*) and W (*CHD1W*). The introns of the *CHD1* genes differ and can determine the sex of avian species [75,76]. As the sex of the 6C2 cell line was unknown, we analyzed our 6C2 RNA Seq data to find that *CHD1W* and other genes on the W chromosome were expressed, showing that the 6C2 was derived from a female chicken.

## 11. Genome Organization of the Genes Involved in Oxygen Transport

Genes involved in the hemoglobin synthesis pathway with molecular functions such as oxygen transporter activity, oxygen binding, and heme binding have extended F_1_ Seq peaks [11,44], demonstrating that these genes were present in chromatin regions that were soluble at physiological ionic strength. The β-globin and α-globin genes are in extended salt-soluble DNase I-sensitive, chromatin domains (33-kb β-globin domain; 60-kb α-globin domain), with the highly acetylated state of the nucleosomes being responsible for this structure [11,23]. The adult β-globin (*HBBA*) and α-globin (*HBA1*) genes are among the genes with the highest transcript levels in the polychromatic erythroid cell. Other genes such as the carbonic anhydrase (*CA2*), ferritin heavy chain 1 (*FTH1*), and transferrin receptor (*TFRC*) are presented as broad peaks in the F_1_ Seq tracks, showing that the chromatin of these genes has a structure that is soluble at physiological ionic strength. Further, the above genes have broad domains of histone PTMs (H3 R2me2s, K4me3, H27ac; H4R3me2a).

The regulatory elements and chromatin features of the β-globin domain are shown in Figure 4 and Figure 5. The epigenomic and transcriptomic data were aligned to the chicken galGal 6 assembly. The β-globin gene is located at the end of chromosome 1 (1q).

When compared to earlier chicken assemblies (galGal 3-5), it is evident that the gene order has been changed at the end of chromosome 1 (q arm) in the galGAl6 assembly. The *POLD3* gene is the last gene at the end of chromosome 1 (q arm) in the galGal 3-5 assemblies, while *ILK* is the last gene at the end of chromosome 1 (q arm) in the galGal 6 assembly. The β-globin genes are located on chromosome 1 (q arm) between the *POLD3* and *ILK* genes. The order of genes on the other arm of chromosome 1 (p arm) is the same for the different galGal assemblies (3 to 6).

The most intense FAIRE Seq peak in the β-globin domain sits at the β-globin *HBBA* promoter region, which is nucleosome-free (Figure 5). Transcription factors binding to this region are Sp1, CAAT, BG1, and NF1-like factor. The *HBBA* promoter is not associated with a CpG island but does have a potential G-quadruplex forming sequence [77]. Next to the nucleosome-free region towards the gene body are nucleosomes modified at H3 (R2me2s, K4me3, K27ac) and H4R3me2a. The *HBBA* enhancer located at the 3′ end of the gene has binding sites for GATA1, NF1-like, AP1/2-like, and CACCC factors. The *HBBA* 3′ enhancer interacts with the *HBBA* promoter in polychromatic erythrocytes [78]. On either side of this enhancer are nucleosomes multiply modified like the nucleosomes at the 5′ end of the *HBBA* gene. As discussed previously, a limitation of the FAIRE Seq method is that it is not suited to detect enhancers. However, there was a depression in the F_1_ Seq peaks at the *HBBA* promoter and enhancer nucleosome-free regions, indicative of a nucleosome-free region.

The locus control region (LCR) plays a critical role in the expression of the *HBBA* gene in polychromatic erythrocytes (Figure 4). The LCR, which is essentially a super-enhancer, has several DNase I hypersensitive sites (HS1–4; nucleosome-free regions) which align with depressions in the F_1_ Seq peaks with two of the hypersensitive regions having a FAIRE Seq peak. The LCR HS4 is categorized as an insulator in that this region sets a boundary between a highly condensed chromatin region and the decondensed β-globin domain chromatin (Figure 5). In polychromatic erythrocytes, the HS4 interacts with the hypersensitive regions next to the OR51M gene (Figure 4) [78]. Within the HS4 is a CpG island; the only CpG island in the entire β-globin domain. Multiple transcription factors including USF factors, CTCF, and VEZF1 (vascular endothelial zinc finger 1) bind to the HS4 region. VEZF1 has an important role in preventing DNA methylation of HS4 [79,80]. VEZF1 is associated with several CpG island promoters in addition to locating with HS4. Poly (ADP-ribose) polymerase 1 (PARP1) also binds to HS4 [81]. Inhibition of PARP1 or mutation of the HS4 PARP1 binding site abolished HS4 barrier activity. The mechanism by which PARP1 is contributing to barrier activity of the HS4 site could be via poly ADP-ribosylation of histones and/or of non-histone chromosomal proteins. There is evidence that PARP1 has a role in the separation of enhancer–promoter interactions [82], suggesting PARP1 has a role in the dynamic juxtapositioning of the LCR with the *HBBA* gene promoter/enhancer [78]. On either side of the HS2 and HS3 regions are stretches of nucleosomes modified at H3 (R2me2s, K27ac) and H4R3me2a. USF1/2 bound at the HS4 recruit protein arginine methyltransferase 1 (PRMT1) which catalyzes H4R3me2a in the surrounding nucleosomes. USF1 also recruits other chromatin-modifying enzymes including the lysine acetyltransferases PCAF and CBP/p300. H4R3me2a also stimulates the activity of the KATs, which would acetylate the core histones, contributing to the solubility of these chromatin regions at physiological ionic strength. Together, the transcription factors binding to the LCR play a key role in recruiting key chromatin-modifying enzymes and shaping the structure of the chromatin domain. The transcription factors loading onto the LCR also have a key role in the interaction between the LCR and the *HBBA* promoter/enhancer and the formation of a transcription hub [78] (Figure 4).

Many transcription factors regulating the expression of the erythroid genes are among the genes with the highest level of transcripts in polychromatic erythrocytes (e.g., *TAL1*, *CTCF*, *NF1A*, *VEZF1*, *Sp3*, *CEBPG* (CCAAT/enhancer binding protein)). However, the role of these and other transcription factors in regulating the expression of genes involved in the chicken hemoglobin synthesis pathway requires further study. For some transcription factor coding genes such as *GATA1* that play critical roles in erythroid gene expression, the genomic location of the gene remains to be identified.

## 12. Innate Immunity

Among the cells participating in the innate immune response in chickens is the erythrocyte [56]. The innate immune system provides a defense of the host against microbes. The chicken mature erythrocyte can induce genes in response to pathogens [5]. Lipopolysaccharides of Gram-negative bacteria or poly(I:C), which is a viral double-stranded RNA mimic, can activate the innate immune response (Figure 6). In response to these agents, mature erythrocytes show increased levels of the transcripts from the *CCL4*, *IFNα*, *TLR3,* and *TLR21* genes. Although the mature erythrocyte is thought to be transcriptionally silent, previously expressed genes maintain a decondensed, poised (poised) chromatin structure [32].

The chicken polychromatic erythrocyte and 6C2 cells express genes that respond to viral RNA as part of the innate immunity machinery [83] (Figure 6). Both cell types express toll-like receptor 3 (*TLR3*), *TLR21*, *TRAF3*, *TRAM1*, *TICAM1*, *TIRAP*, *MYD88*, *IRF7*, and *IKBKB* (inhibitor of nuclear factor kappa B kinase subunit beta), and *NFKB2*. Chicken erythroleukemic 6C2 cells, but not polychromatic erythrocytes, have transcripts for *TLR4* and *TLR21*; however, the chromatin of both genes has active marks associated with the 5′ gene, suggesting that these genes are poised for transcription in the polychromatic erythrocyte.

The *TLR3* gene responds to double-stranded RNA produced by viral infection. Incubation of chicken polychromatic erythrocytes with the double-stranded RNA mimic, poly(I:C), results in increased *TLR3* transcript levels [63]. In the chicken polychromatic erythrocyte, the 5′ region of the *TLR3* gene has F_1_ Seq peaks indicative of enhanced chromatin solubility at physiological ionic strength and nucleosomes that are multiply modified (H3R2me2s, K4me3, K27ac, and H4R3me2a). At the 3′end of the gene in the intergenic region is a CpG island that has a FAIRE Seq peak (nucleosome-free region), surrounded by nucleosomes that are modified at H3R2me2s, K27ac, and H4R3me2a), which is a signature associated with enhancers. Whether this is an enhancer for the *TLR3* gene remains to be determined.

Chromosome Z has a cluster of four *IFNA3* genes and *INFW1*. Transcripts for these *IFN* genes were not detected in our RNA Seq data for chicken polychromatic erythrocyte and 6C2 cells. However, the chromatin structure of this region in chicken polychromatic erythrocytes appears as a gene region poised for transcription. The 5′ end of the *IFN* cluster has F_1_ Seq peaks and a salt-soluble chromatin structure, and the nucleosomes throughout the cluster have multiple modifications for H3 (R3me2s, K4me3, K27ac) and H4R3me2a. This chromatin region is unstable as indicated by the FAIRE Seq peaks (nucleosome-free regions) throughout the cluster. It is noted that the cluster has many CpG islands.

IRF1 (interferon regulatory factor) and NF-κB are involved in the expression of the *IFNA3* and *INFW1* genes [84]. The chicken polychromatic erythrocyte *IRF1* gene is immersed in F_1_ Seq broad peaks (salt soluble domain), and the gene promoter region and body have nucleosomes that are multiply modified. In addition to *IRF1*, the chicken polychromatic erythrocyte, and 6C2 cells express several other interferon regulatory factors including *IRF2*, *IRF8*, and *IRF9* (also known as *IRF10*). All expressed *IRF* genes, except *IRF8*, are associated with CpG islands.

Among the interleukins, 6C2 and polychromatic erythrocytes have transcripts for *IL-15*, but not other interleukins. The chromatin structure of the *IL-15* gene in polychromatic erythrocytes is an enigma as the gene and surrounding regions do not have any distinguishing features that are associated with gene transcription. In contrast, *IL1B* transcripts were not detected in chicken polychromatic erythrocytes but the gene has chromatin features of a transcribed gene. Similarly, transcripts from the cytokine *CCL4* gene were not detected in polychromatic erythrocytes but the gene chromatin has many features of a transcribed gene. It appears that both the *IL1B* and *CCL4* genes are poised for expression. Consistent with this observation, *IL1B* transcripts were observed for 3 h following stimulation of chicken polychromatic erythrocytes with poly(I:C) [63].

## 13. Concluding Remarks

The nucleated erythrocyte in chickens has a critical role in carrying oxygen to tissues and the removal of carbon dioxide. The erythrocyte also contributes to protecting the host from various pathogens. The configuration of the erythrocyte epigenome plays a major role in the organization and function of genes involved in these red blood cell functions. Transcription factors, chromatin modifying enzymes, and histone PTMs play a critical role in maintaining a transcription-permissive decondensed chromatin state in a background of highly condensed chromatin.

The erythrocyte epigenome is responsive to both internal (metabolism) and external (stress, social environment) conditions that impact chromatin organization and gene expression. The order of genes in the chicken genome is like that of the human genome but the chicken has a smaller genome. Thus, the chicken red blood cell is an excellent model system to study the consequences of the environment on the epigenome and translate these insights to human health.

There is still much to be done in the characterization of the chicken erythrocyte epigenome and the proteins and chromatin-modifying enzymes involved in the writing, reading, and removal of the epigenomic marks. The chicken erythrocyte chromatin fractionation protocol can be applied to study novel histone PTMs and obtain insights as to whether the PTM contributes to gene expression or repression when antibodies to that histone PTM are not available. For example, we know very little about the acid labile H4His18ph [85] and histone variant-specific PTMs [86]. The ability to isolate a highly enriched transcriptionally active chromatin presents opportunities to study atypical nucleosome structures associated with this chromatin [33]. Further, mass spectrometry analyses of the proteins associated with transcriptionally active chromatin will document the chromatin modifying enzymes, epigenetic readers, transcription factors, and other factors involved in maintaining the decondensed transcription-ready chromatin state. The smaller chicken genome leads itself in mapping histone PTMs, variants, chromatin modifying enzymes, etc. However, it is also important to understand which histone PTMs/variants coexist in a single nucleosome. Characterization of the histone variant and PTM composition of single nucleosomes associated with transcribing chromatin is made possible by combining the isolation of transcriptionally active chromatin with immunoprecipitation of mononucleosomes carrying specific histone PTMs and/or variants [87]. Aided by a powerful chromatin fractionation protocol and new emerging epigenomic approaches, studies with the chicken erythrocyte model system will continue to make seminal contributions to our understanding of the regulation of gene expression.

## Figures and Tables

**Figure 1 ijms-24-08287-f001:**
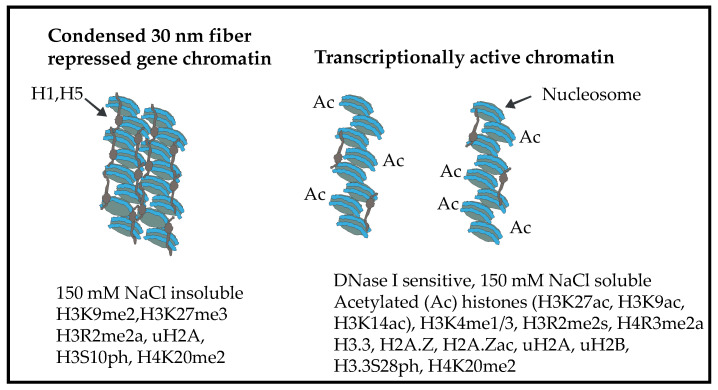
Repressed and transcriptionally active chromatin. In chicken polychromatic erythrocytes, repressed chromatin on the left has a condensed 30 nm chromatin structure and is insoluble at physiological ionic strength. Transcriptionally active chromatin shown on the right has a decondensed chromatin structure that is soluble at physiological ionic strength. Histone acetylation plays a key role in keeping the chromatin fibers separated. Histone PTMs and variants associated with each chromatin state are listed.

**Figure 2 ijms-24-08287-f002:**
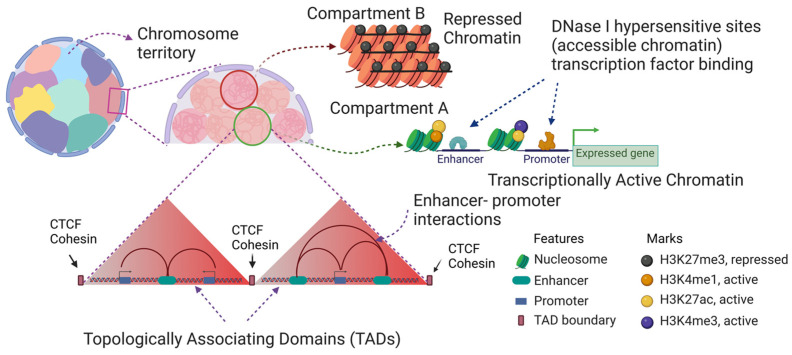
Representation of the typical chromatin structure in vertebrates. The illustration shows chromosome territories, compartments, and topologically associating domains, which chicken erythrocytes do not have. Compartment B has repressed chromatin and repressive histones marks, while compartment A has transcribed chromatin and active marks. Created with BioRender.com (accessed on 24 March 2023).

**Figure 3 ijms-24-08287-f003:**
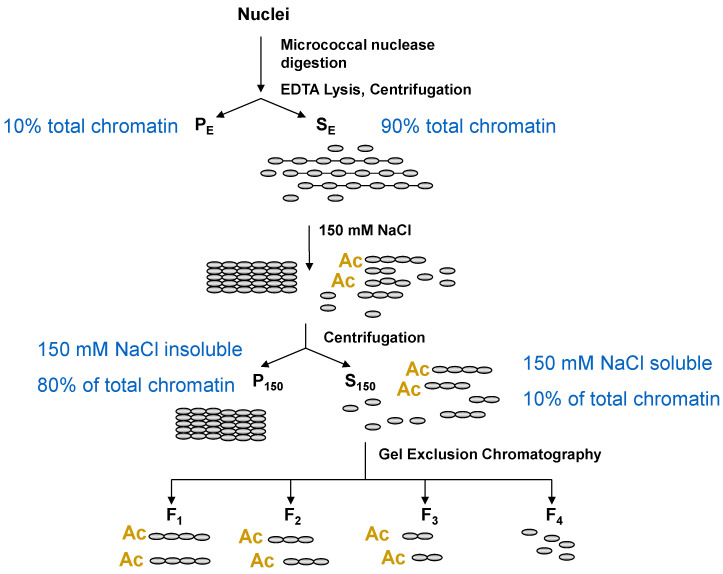
Chromatin fractionation schema. For further details of the chromatin fractionation protocol, see [12]. Transcriptionally active/poised chromatin is enriched in fractions F_1_, F_2_, and F_3_ which contain highly acetylated core histones. The P_150_ chromatin fraction has transcriptionally repressed chromatin. The P_E_ fraction has the residual nuclear material including the nuclear matrix, and transcriptionally active/poised and repressed chromatin. Ac, acetylated histone.

**Figure 4 ijms-24-08287-f004:**
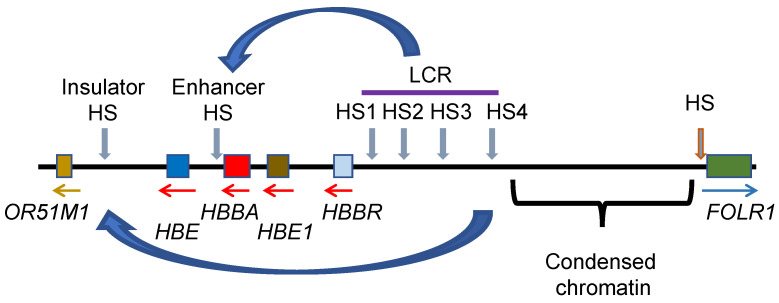
The β-globin domain in chicken erythrocytes is shown. LCR, locus control region, HS, hypersensitive site (nucleosome-free region). The arrows indicate the direction of transcription. The curved arrows show the interactions between the LCR and the *HBBA* enhancer and insulator element.

**Figure 5 ijms-24-08287-f005:**
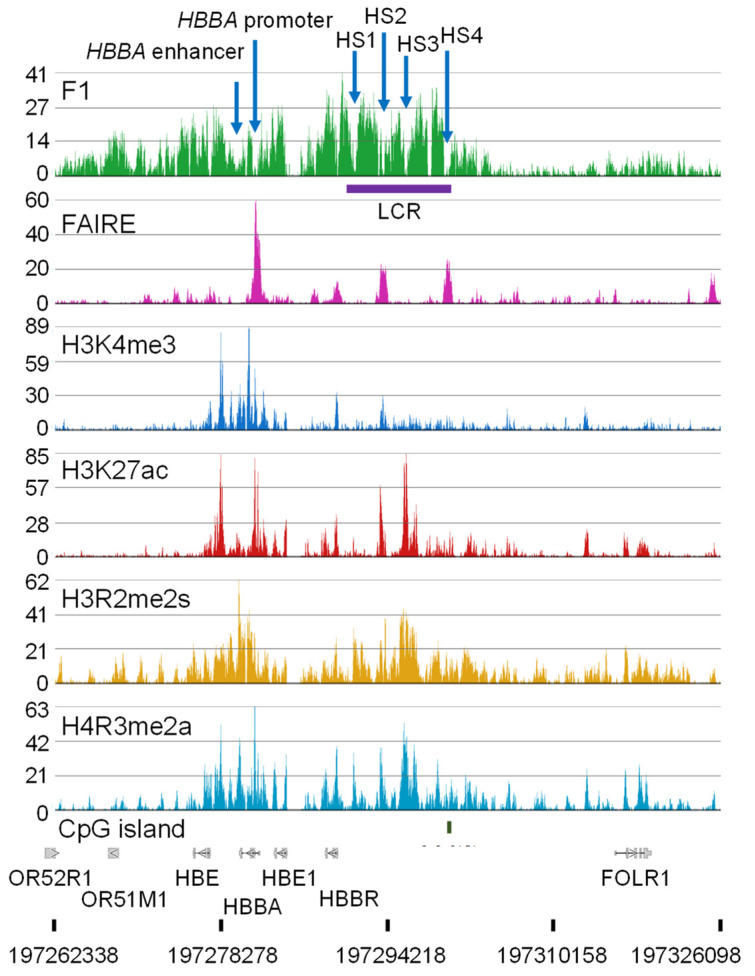
The β-globin domain in chicken polychromatic erythrocytes. Shown are the tracks for F_1_ Seq, FAIRE Seq, and ChIP Seq for H3K4me3, H3K27ac, H3R2me2s, and H4R3me2a.

**Figure 6 ijms-24-08287-f006:**
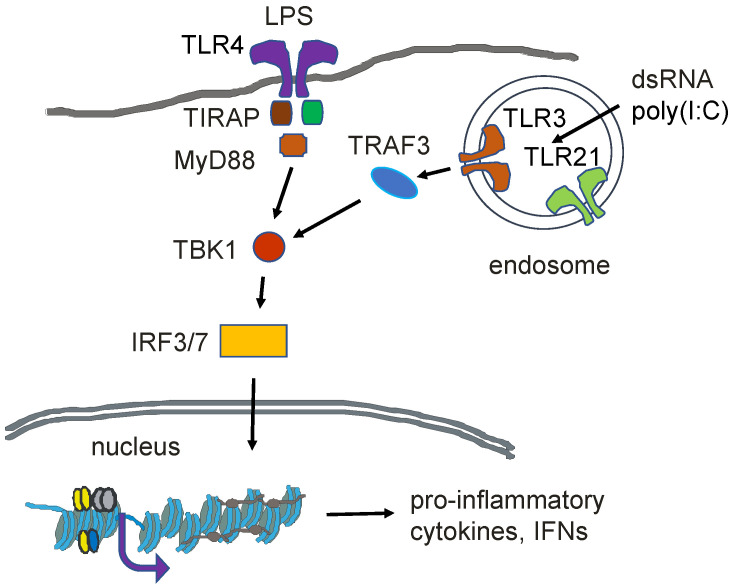
Innate immune response in chicken erythrocytes. Activation of the TLR3 and TLR4 receptors by dsRNA and LPS, respectively, initiates a series of modification and protein binding events resulting in the expression of pro-inflammatory cytokines and interferon (IFN).

## Data Availability

The study was done with publicly available datasets which can be obtained through Gene Expression Omnibus.

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
