# Peer review of "Chicken Erythrocyte: Epigenomic Regulation of Gene Activity"

_ijms, 2023, doi:10.3390/ijms24098287_

Round 1
Reviewer 1 Report
The manusript is well written and knowledgeable and should be published without revision. The review is quite comprehensive and novel given the fact that it reviews the up-to-date information about epigenomic regulation of gene activity in chicken erythrocyte. Most studies focus on human genome, however, the current study picks chicken and it would provide distinctive knowledge for the scientific community. Plus, the review is very knowledgeable and very well written, and that's why I suggested it to be published without further revision.
Author Response
We are pleased that reviewer one enjoyed reading the review and wrote in the review that “the review is very knowledgeable and very well written”.
Reviewer 2 Report
Overall, this research article represents a fascinating exploration on “Chicken erythrocyte: Epigenomic Regulation of Gene Activity”
Abstract: The abstract provides a clear and concise overview of the research topic, including the relevance of the chicken erythrocyte as a model organism for studying epigenetics and its physiological functions. The objectives of the study and the methods used are presented, and the abstract ends with a statement on the epigenomic features that allow for erythroid gene expression in a repressive chromatin background. However, it would be useful to include some results or findings in the abstract to make it more informative.
Introduction: The introduction is well-written and provides essential background information on the chicken genome, including its similarities and differences with the human genome. The physiological functions of the chicken erythrocyte are clearly explained, and the role of epigenetics in regulating gene activity is introduced. The purpose of the review is also stated explicitly.
Chromatin structure of the chicken erythrocyte: This section is informative and provides details on the chromatin structure of the chicken erythrocyte. The methods used to study the chromatin structure are briefly described, and the results are presented clearly. However, some of the technical terms used may not be familiar to readers who are not experts in the field, so it would be helpful to provide explanations or definitions.
Chicken erythrocyte promoters and DNA Methylation: text presents a well-written and informative review of the characteristics and regulation of chicken promoters and DNA methylation. The author provides detailed explanations of various concepts and techniques used in the study of these genomic features, making the text accessible to readers with varying levels of knowledge on the subject.
Chromatin fractionation: chromatin fractionation presents a detailed description of a powerful method to study chromatin composition in chicken erythrocytes. The authors provide a clear rationale for why these cells are well-suited for fractionation and isolation of transcriptionally active/poised chromatin, which is important for investigating the chromatin composition of active versus repressed genes. The authors also provide a helpful figure (Figure 1) to illustrate the chromatin fractionation schema and the location of transcriptionally active/poised and repressed chromatin fractions. the section is well-written and informative. One suggestion for improvement would be to include more details on how the fractionation was performed (e.g., buffer compositions, centrifugation conditions), as this would be helpful for researchers looking to replicate the method.
Histone acetylation, chromatin solubility, and the DNase I sensitive chromatin domains: This provides a good overview of the early studies on histone acetylation and chromatin solubility in chicken erythrocytes. However, here are some suggestions to improve the clarity and readability of the text:
1. Use simpler language: The article contains many technical terms that may be difficult for non-experts to understand. To improve readability, it is recommended to use simpler language wherever possible.
2. Clarify unclear statements: There are a few statements in the article that are unclear or could be misleading. For example, in the sentence "In polychromatic erythrocytes, H4 is acetylated at a rate with a half-life of 12 min, while in mature erythrocytes, H4 is acetylated at two rates (t½ of 12 min and 300 min)", it is not clear what is meant by "two rates". Clarifying statements like this can improve the overall clarity of the article.
3. Provide more recent research: While the article provides a good overview of early research on histone acetylation and chromatin solubility, it would be helpful to include more recent research to give the reader a sense of how this field has progressed since these early studies.
Chicken erythrocyte histone genes and variants: the section provides a detailed description of the expression and regulation of histone genes in chicken polychromatic erythrocytes, with a focus on the H1F0 gene. The use of references and specific experimental results lends credibility to the information presented. However, there are a few suggestions for improvement:
- Clarify the significance of the findings. While the section provides a lot of detailed information about the expression and regulation of histone genes in chicken erythrocytes, it is not entirely clear why this is important or what the broader implications are. Providing a clearer statement about the significance of the findings would make the section more impactful.
- Use simpler language where possible. The section uses a lot of technical terms and abbreviations, which may be difficult for readers without a background in the field to understand. Simplifying the language where possible or providing definitions for technical terms would make the section more accessible to a wider audience.
Chicken histone PTMs, nucleosome-free regions, and genomic mapping:. The language used in this section is scientific and specific. However, there are a few areas where the writing could be improved. Here are some suggestions:
1. The first sentence is a bit unclear. Consider rephrasing it to make it more concise and easier to understand. For example, "We characterized transcribed genes in chicken polychromatic erythrocytes' euchromatin/compartment A by profiling several histone PTMs at the promoter, enhancer, and gene body of individual genes using F1 Seq, chromatin immunoprecipitation (ChIP) Seq, RNA Seq, and FAIRE Seq to build an epigenomic map."
2. In sentence 348, consider specifying which genome assembly was used for sequence alignments.
3. In sentence 349, consider clarifying how the histone PTMs were mapped out using the sequencing data.
4. In sentence 351, consider defining what super-enhancers are for readers who may not be familiar with the term.
5. In sentence 352, consider providing more context on why mapping H3R2me2s and H4R3me2a is significant.
6. In sentence 355, consider rephrasing to clarify what ChIP qPCR is and how it differs from ChIP Seq.
Broad histone PTM domains and chromatin remodeling by CHD1: This section is well-written and informative. Here are a few suggestions:
1. Clarify the role of CHD1 in the chromatin remodeling of the broad histone PTM domains. It would be helpful to provide more information on how CHD1 promotes nucleosome assembling, remodeling, sliding, and promotes their regular spacing.
2. Provide more information on the limitations of the study and potential areas for future research. This would help readers understand the scope of the research and the potential for further investigation.
Genome organization of the genes involved in oxygen transport: This passage provides a detailed description of the genome organization of genes involved in oxygen transport, particularly hemoglobin synthesis pathway genes. It describes the location of these genes within chromatin domains, their molecular functions, and the chromatin features associated with them. some clarification might be helpful regarding the use of technical terminology and abbreviations. For example, it may be useful to define or explain terms such as "extended F1 Seq peaks" and "hypersensitive site," especially for readers who may not be familiar with these concepts. Overall, providing more context and clarification would help make this passage more accessible and informative to a wider range of readers.
Innate immunity: This section provides detailed information about the innate immune response in chickens and the role of erythrocytes in this process. However, it may be helpful to provide more background information for readers who are not familiar with the specific genes and mechanisms discussed. Additionally, the passage could be organized more clearly to help readers follow the flow of information.
Conclusion: The conclusion of this manuscript provides a concise summary of the main findings and the potential of using the chicken erythrocyte as a model system for studying the epigenetic regulation of gene expression. The use of abbreviations is helpful for readers to follow the text, and the author contributions, funding, institutional review board statement, informed consent statement, data availability statement, acknowledgments, and conflicts of interest are all appropriate and well-written.
However, there are a few areas where the conclusion could be improved. Firstly, the author could include some suggestions for future research directions or specific questions that need to be addressed to advance the field. This would help to stimulate further research and indicate areas that need more attention. Additionally, the author could provide a brief summary of the main implications of the research for human health or disease. Finally, the conclusion could be written in a more active voice to engage readers and emphasize the significance of the findings.
Overall, the paper is well-written and provides valuable information on the epigenomic regulation of gene activity in the chicken erythrocyte. The research is relevant and informative, and the paper is likely to be of interest to researchers in the field of epigenetics. Some minor revisions could be made to make the paper more accessible to non-experts in the field, but overall, the paper is well-executed.
Author Response
Overall response: We appreciate that the reviewer critiqued the review in detail. In responding to the reviewer’s request for greater clarity in various sections of the review, we have added new figures which will aid the reader in understanding the text.
Comment 1.
Overall, this research article represents a fascinating exploration on “Chicken erythrocyte: Epigenomic Regulation of Gene Activity”
Abstract: The abstract provides a clear and concise overview of the research topic, including the relevance of the chicken erythrocyte as a model organism for studying epigenetics and its physiological functions. The objectives of the study and the methods used are presented, and the abstract ends with a statement on the epigenomic features that allow for erythroid gene expression in a repressive chromatin background. However, it would be useful to include some results or findings in the abstract to make it more informative.
Response: In agreement with reviewer one, we are OK with the abstract as written. As this is a review and not an original article, we are of the opinion that stating results is not what one would put in an abstract of a review.
Comment 2.
Introduction: The introduction is well-written and provides essential background information on the chicken genome, including its similarities and differences with the human genome. The physiological functions of the chicken erythrocyte are clearly explained, and the role of epigenetics in regulating gene activity is introduced. The purpose of the review is also stated explicitly.
Chromatin structure of the chicken erythrocyte: This section is informative and provides details on the chromatin structure of the chicken erythrocyte. The methods used to study the chromatin structure are briefly described, and the results are presented clearly. However, some of the technical terms used may not be familiar to readers who are not experts in the field, so it would be helpful to provide explanations or definitions.
Chicken erythrocyte promoters and DNA Methylation: text presents a well-written and informative review of the characteristics and regulation of chicken promoters and DNA methylation. The author provides detailed explanations of various concepts and techniques used in the study of these genomic features, making the text accessible to readers with varying levels of knowledge on the subject.
Chromatin fractionation: chromatin fractionation presents a detailed description of a powerful method to study chromatin composition in chicken erythrocytes. The authors provide a clear rationale for why these cells are well-suited for fractionation and isolation of transcriptionally active/poised chromatin, which is important for investigating the chromatin composition of active versus repressed genes. The authors also provide a helpful figure (Figure 1) to illustrate the chromatin fractionation schema and the location of transcriptionally active/poised and repressed chromatin fractions. the section is well-written and informative. One suggestion for improvement would be to include more details on how the fractionation was performed (e.g., buffer compositions, centrifugation conditions), as this would be helpful for researchers looking to replicate the method.
Response: We added new figures (Figure 1 (line 69) and Figure 2 (line 90)) to aid in the understanding of various aspects of epigenetics/chromatin structure presented in the text. For the chromatin fractionation protocol details, we added a sentence on line 217 that lists references that provide experimental details.
Comment 3.
Histone acetylation, chromatin solubility, and the DNase I sensitive chromatin domains: This provides a good overview of the early studies on histone acetylation and chromatin solubility in chicken erythrocytes. However, here are some suggestions to improve the clarity and readability of the text:
- Use simpler language: The article contains many technical terms that may be difficult for non-experts to understand. To improve readability, it is recommended to use simpler language wherever possible.
Response: We added new figures (Figure 1 and 2) to aid in understanding this section.
- Clarify unclear statements: There are a few statements in the article that are unclear or could be misleading. For example, in the sentence "In polychromatic erythrocytes, H4 is acetylated at a rate with a half-life of 12 min, while in mature erythrocytes, H4 is acetylated at two rates (t½ of 12 min and 300 min)", it is not clear what is meant by "two rates". Clarifying statements like this can improve the overall clarity of the article.
Response: We added the sentence (line 283) “The studies done by Zhang and Nelson demonstrate that core histones associated with transcriptionally active chromatin are rapidly acetylated and deacetylated” which pretty much summarizes the text before going into details about the rates of acetylation and deacetylation.
- Provide more recent research: While the article provides a good overview of early research on histone acetylation and chromatin solubility, it would be helpful to include more recent research to give the reader a sense of how this field has progressed since these early studies.
Response: We have presented the latest results on histone acetylation with chicken erythrocytes.
Comment 4.
Chicken erythrocyte histone genes and variants: the section provides a detailed description of the expression and regulation of histone genes in chicken polychromatic erythrocytes, with a focus on the H1F0 gene. The use of references and specific experimental results lends credibility to the information presented. However, there are a few suggestions for improvement:
- Clarify the significance of the findings. While the section provides a lot of detailed information about the expression and regulation of histone genes in chicken erythrocytes, it is not entirely clear why this is important or what the broader implications are. Providing a clearer statement about the significance of the findings would make the section more impactful.
Response: The following sentence was added on line 340 “Other than the globin genes, the erythroid regulation of the H1F0 gene expression has been extensively characterized and provides an example of regulatory elements and factors involved in erythroid gene expression”.
- Use simpler language where possible. The section uses a lot of technical terms and abbreviations, which may be difficult for readers without a background in the field to understand. Simplifying the language where possible or providing definitions for technical terms would make the section more accessible to a wider audience.
Response: We added figures (Figures 1 and 2) to help with this understanding.
Comment 5.
Chicken histone PTMs, nucleosome-free regions, and genomic mapping:. The language used in this section is scientific and specific. However, there are a few areas where the writing could be improved. Here are some suggestions:
- The first sentence is a bit unclear. Consider rephrasing it to make it more concise and easier to understand. For example, "We characterized transcribed genes in chicken polychromatic erythrocytes' euchromatin/compartment A by profiling several histone PTMs at the promoter, enhancer, and gene body of individual genes using F1 Seq, chromatin immunoprecipitation (ChIP) Seq, RNA Seq, and FAIRE Seq to build an epigenomic map."
Response: The first sentence was revised as suggested (see line 368).
- In sentence 348, consider specifying which genome assembly was used for sequence alignments.
Response: On line 376 we specify the genome assembly used for sequence alignments.
- In sentence 349, consider clarifying how the histone PTMs were mapped out using the sequencing data.
Response: On line 372 we added the following “Histone PTMs were located in the chick erythroid chromatin by using ChIP Seq which involves immunoprecipitation of fragmented formaldehyde cross-linked chromatin with antibodies specific to a histone PTM. The transcriptome of the erythroid cell is determined by RNA Seq which is the sequencing of cellular RNA. FAIRE Seq as we presented previously maps genomic regions that are nucleosome free”.
- In sentence 351, consider defining what super-enhancers are for readers who may not be familiar with the term.
Response: On line 382, we added “super-enhancers, which are a group of enhancers in close genomic proximity”.
- In sentence 352, consider providing more context on why mapping H3R2me2s and H4R3me2a is significant.
Response: On line 384, we added “Our study showed that these two marks are often together and locate in regulatory regions of the chicken erythroid genome”.
- In sentence 355, consider rephrasing to clarify what ChIP qPCR is and how it differs from ChIP Seq.
Response: On line 387 we added “ChIP qPCR, which analyzes the immunoprecipitated DNA using PCR rather than next-generational DNA sequencing”.
Comment 6.
Broad histone PTM domains and chromatin remodeling by CHD1: This section is well-written and informative. Here are a few suggestions:
- Clarify the role of CHD1 in the chromatin remodeling of the broad histone PTM domains. It would be helpful to provide more information on how CHD1 promotes nucleosome assembling, remodeling, sliding, and promotes their regular spacing.
Response: On line 404 we added “For further information on the mechanism of CHD1 remodeling see [65,66]”.
- Provide more information on the limitations of the study and potential areas for future research. This would help readers understand the scope of the research and the potential for further investigation.
Response: On line 415, we added “It is acknowledged, however, that there are other CHD chromatin remodelers that may be involved in nucleosome remodeling of broad histone PTM domains [72]”.
Comment 7.
Genome organization of the genes involved in oxygen transport: This passage provides a detailed description of the genome organization of genes involved in oxygen transport, particularly hemoglobin synthesis pathway genes. It describes the location of these genes within chromatin domains, their molecular functions, and the chromatin features associated with them. some clarification might be helpful regarding the use of technical terminology and abbreviations. For example, it may be useful to define or explain terms such as "extended F1 Seq peaks" and "hypersensitive site," especially for readers who may not be familiar with these concepts. Overall, providing more context and clarification would help make this passage more accessible and informative to a wider range of readers.
Response: We thought it was helpful to show the two parts of this figure separately (now Figures 4 and 5). For clarification of F1 Seq peaks we added on line 433 “….F1 Seq tracks, showing that the chromatin of these genes has a structure that is soluble at physiological ionic strength”. At the beginning of this section (line 424) we stated that “had extended F1 Seq peaks [27,39], demonstrating that these genes were present in chromatin regions that were soluble at physiological ionic strength”.
We added to the legend of Figure 4 (line 441) that DNase hypersensitive site is nucleosome free.
Comment 8.
Innate immunity: This section provides detailed information about the innate immune response in chickens and the role of erythrocytes in this process. However, it may be helpful to provide more background information for readers who are not familiar with the specific genes and mechanisms discussed. Additionally, the passage could be organized more clearly to help readers follow the flow of information.
Response: On line 535, we added “The innate immune system provides a defense of the host against microbes. The chicken mature erythrocyte can induce genes in response to pathogens [4]. Lipopolysaccharide of Gram-negative bacteria or poly(I:C), which is a viral double-stranded RNA mimic, can activate the innate immune response (Figure 6)”. The Figure showing the pathway (Figure 6, line 542) was revised to aid in understanding the text.
Comment 9.
Conclusion: The conclusion of this manuscript provides a concise summary of the main findings and the potential of using the chicken erythrocyte as a model system for studying the epigenetic regulation of gene expression. The use of abbreviations is helpful for readers to follow the text, and the author contributions, funding, institutional review board statement, informed consent statement, data availability statement, acknowledgments, and conflicts of interest are all appropriate and well-written.
However, there are a few areas where the conclusion could be improved. Firstly, the author could include some suggestions for future research directions or specific questions that need to be addressed to advance the field. This would help to stimulate further research and indicate areas that need more attention. Additionally, the author could provide a brief summary of the main implications of the research for human health or disease. Finally, the conclusion could be written in a more active voice to engage readers and emphasize the significance of the findings.
Overall, the paper is well-written and provides valuable information on the epigenomic regulation of gene activity in the chicken erythrocyte. The research is relevant and informative, and the paper is likely to be of interest to researchers in the field of epigenetics. Some minor revisions could be made to make the paper more accessible to non-experts in the field, but overall, the paper is well-executed.
Response: The last paragraph starting on line 607 of the Conclusion section has been revised in providing the reader more insights into future plans in using the chicken erythrocyte system to understand the regulation of gene expression. As this review was part of a special issue on erythrocytes, we kept the focus as such. In writing the revised paragraph we used the active voice.
Reviewer 3 Report
Please see the attachment.

Author Response
We thank the reviewer for catching these errors in the manuscript.
Response to reviewer 3 comments (Note revised text is highlighted)
- For Figure 1 on pg 3, it would be good to give some context and details regarding how was it determined that differential solubility in 150 mM NaCl corresponded specifically to condensed 30 nm fiber repressed chromatin vs. transcriptionally active chromatin.
Response: On page 2, line 82, we added the following text.
The highly acetylated state of transcriptionally active/poised chromatin prevents H1/H5-induced chromatin condensation (30 nm fiber) and insolubility at 150 mM NaCl [7]. Incubating chicken polychromatic erythrocytes in the absence of a histone deacetylase inhibitor (sodium butyrate) resulted in the deacetylation of transcriptionally active/poised chromatin. In the deacetylated state, the transcriptionally active/poised chromatin fragments underwent H1/H5-induced condensation and salt-insolubility, demonstrating the importance of histone acetylation in maintaining the decondensed state of transcription-ally active/poised chromatin. These studies used labeled DNA probes to transcription active/poised (e.g., HBBA, H1FO) and repressed (VTG1) gene sequences in Southern blot experiments, PCR with primers to active or repressed DNA sequences, and next-generation DNA sequencing to determine the structure (e.g., DNase I sensitivity) and salt solubility properties of active/poised and repressed gene chromatin [8–10].
- On pg 5, line 171, should there be an “and” between “chromatin” and “by”?
Response: The sentence on page 4, line 172 was revised as follows.
The gene-silencing function of DNA methylation is achieved by the recruitment of histone deacetylase (HDAC) complexes that have methyl-DNA binding motifs, and through relationships between lysine methyltransferases (Suppressor of Variegation 3-9 Homolog 1/2, SUV39h ½, G9) and DNA methylation [25].
- Figure 3 appeared twice on pgs 7 and 8 and needed to be in higher resolution. Moreover, the figure as drawn was confusing and more details are needed in the figure legend. First, it is unclear what the “10% soluble” is referring to: is it that 10% of the chromatin in SE fractionates to the S150 fraction and 90% goes to the P150 fraction? Second, the figure, as drawn, seemed to suggest that only the soluble/active chromatin was centrifuged to give rise to the S150 and P150 fractions. I understand that is not how the procedure was performed, but the figure was unclear. Third, there were only mono-nucleosomes depicted in the S150 fraction, which is confusing. Lastly, it would be better to put the gel exclusion chromatography arrow directly under the S150 fraction to clarify that the F1 – 4 fractions came from the S150 fraction (not the S150 + P150).
Response: We revised Figure 3 to address this comment.
- Pg 6, lines 222 to 224: I think more explanation needs to be given to explain how “gel exclusion chromatography helped to segregate active/euchromatin from heterochromatin”. My understanding is that the gel exclusion was separating chromatin fragments based on size, so how does the size of the oligonucleosomes arrays vs. single nucleosomes separate euchromatin vs. heterochromatin? Especially when it was noted that highly active genes were found in the F1 (larger oligonucleosome array) fraction?
Response: We have written a more detailed description of the fractionation procedure which addresses this comment. On page 5, line 222, we wrote the following.
The first steps in the chromatin fractionation protocol include micrococcal nuclease digestion of chromatin and lysis of the nuclei. Under these conditions, chromatin fibers decondense and are soluble. Following centrifugation, most chromatin fragments (active/poised and repressed DNA) are present in the supernatant (fraction SE), while the pellet (fraction PE) has some chromatin (both transcriptionally active and repressed genes) that is bound to the insoluble nuclear material which includes the nuclear lamina and nuclear matrix. The addition of NaCl (to 150 mM) to the SE fraction renders most chromatin fragments in-soluble. The insoluble chromatin fragments form higher-order chromatin structures such as the 30 nm fiber. The insoluble chromatin fragments are collected by centrifugation, yielding fraction P150. The supernatant has salt-soluble chromatin fragments (polynucleosomes, oligonucleosomes, and mononucleosomes) (fraction S150). The salt-soluble polynucleosomes and oligonucleosomes are enriched in active/poised DNA sequences. The mononucleosomes are derived from active/poised and repressed chromatin. Gel exclusion chromatography is applied to separate the longer chromatin fragments (fractions F1-3) from the mononucleosomes (fraction F4). As shown in Figure 3, transcriptionally active/poised chromatin is present in fractions S150, F1-3, and PE, while repressed chromatin is in fractions P150 and PE.
- Pg 7, line 239: the authors should clarify which “elongation” process is being referred to (transcription elongation?), and also add some explanation of the Ushaped nucleosomes. It was noted that “polynucleosomes” present in the F1 fraction have U-shaped nucleosomes – do the authors mean that all nucleosomes in that fraction are U-shaped? Are those U-shaped nucleosomes more resistant to micrococcal nuclease digestion given the F1 fraction consists of larger oligonucleosome arrays (i.e. protected from MNase digestion)?
Response: On page 6, line 243, we revised the text as follows:
Polynucleosomes present in the transcriptionally active/poised chromatin fraction F1 have canonical nucleosomes and atypical nucleosomes that are described as U-shaped nucleosomes [32].
Also added on page 6, line 246
The presence of these U-shaped nucleosomes in the polynucleosomes suggests that these atypical nucleosomes are not overly sensitive to micrococcal nuclease digestion.
- Pg 8, line 271: the statement “an H2A-H2B dimer with H2A.Z had increase stability” is oddly worded – do the authors mean “H2A.Z-H2B dimers are more stable than H2A-H2B dimers”?
Response: The sentence now on page 7, line 275 was revised as follows.
In contrast, the interaction of the H2A.Z with the (H3-H4)2 tetramer and/or nucleosomal DNA was stronger than that of histone H2A [40].
- Pg 8, lines 273/274: the authors should give more details and context to explain the statement “chromatin-associated newly synthesized H2A and H2B were prominently ubiquitinated”. How was this determined? How were the ubiquitinated H2A/H2B detected and how were they identified as newly synthesized?
Response: On page 7, line 279 we added the following text.
In this study, chicken polychromatic erythrocytes were labeled with L-[4,5-3H] lysine for one hour to label newly synthesized histones [41]. The labeled histones from unfractionated chromatin and chromatin fractions (SE, S150, P150, PE) were resolved by two-dimensional electrophoresis (acetic acid-urea-Triton X-100 (AUT) polyacrylamide gels to SDS polyacrylamide gel electrophoresis (PAGE)) followed by fluorography. AUT-PAGE resolves histones by size, charge, and hydrophobicity [42]. Ubiquitinated H2A and H2B have distinctive positions on the one-dimensional AUT and two-dimensional gel patterns. The fluorograms showed that uH2A and uH2B in the transcriptionally active/poised-enriched chromatin fraction F1 were labeled at levels comparable to or greater than the parent histone.
Figs 4 and 6 on pgs 12 and 15 respectively were duplicated and overlapping on the PDF file supplied for review.
Response: This duplication was a problem in converting the word doc with track changes to a PDF.
- There are 2 columns of numbers of the left-hand side of Fig 5 that should be cropped or fixed.
Response: This problem has been corrected.
Round 2
Reviewer 2 Report
The authors have fully addressed my concerns.
Author Response
We are pleased that we have fully addressed the reviewer's concerns which have improved the manuscript.